# Deacylated Derivative of Hericenone C Treated by Lipase Shows Enhanced Neuroprotective Properties Compared to Its Parent Compound

**DOI:** 10.3390/molecules28114549

**Published:** 2023-06-05

**Authors:** Sonam Tamrakar, Dongmei Wang, Eri Hiraki, Chunguang Han, Yang Ruan, Ahmed E. Allam, Yhiya Amen, Yoshinori Katakura, Kuniyoshi Shimizu

**Affiliations:** 1Department of Agro-Environmental Sciences, Faculty of Agriculture, Kyushu University, Fukuoka 819-0395, Japan; tamraka3@msu.edu (S.T.); spswdm@hotmail.com (D.W.); e.hiraki.835@gmail.com (E.H.); syunkouk@gmail.com (C.H.); ahxcry9619@hotmail.com (Y.R.); ahmedezzaldeanallam@gmail.com (A.E.A.); 2Department of Fisheries and Wildlife, College of Agriculture and Natural Resources, Michigan State University, East Lansing, MI 48824, USA; 3Department of Pharmacognosy, Faculty of Pharmacy, Al-Azhar University, Assiut 71524, Egypt; 4Department of Pharmacognosy, Faculty of Pharmacy, Mansoura University, Mansoura 35516, Egypt; yhiyaamen@gmail.com; 5Department of Genetic Resources Technology, Faculty of Agriculture, Kyushu University, Fukuoka 819-0395, Japan; katakura@grt.kyushu-u.ac.jp

**Keywords:** *Hericium erinaceus*, enzyme digestion, deacylhericenone, neurotrophic factors, BDNF, oxidative stress

## Abstract

*Hericium erinaceus*, a mushroom species commonly known as Yamabushitake in Japan, is known to have a stimulatory effect on neurotrophic factors, such as brain-derived neurotrophic factor (BDNF) and nerve growth factor (NGF). Hericenone C, a meroterpenoid with palmitic acid as the fatty acid side chain, is reported to be one such stimulant. However, according to the structure of the compound, the fatty acid side chain seems highly susceptible to lipase decomposition, under in vivo metabolic conditions. To study this phenomenon, hericenone C from the ethanol extract of the fruiting body was subjected to lipase enzyme treatment and observed for changes in the chemical structure. The compound formed after the lipase enzyme digestion was isolated and identified using LC-QTOF-MS combined with ^1^H-NMR analysis. It was found to be a derivative of hericenone C without its fatty acid side chain and was named deacylhericenone. Interestingly, a comparative investigation of the neuroprotective properties of hericenone C and deacylhericenone showed that the BDNF mRNA expression in human astrocytoma cells (1321N1) and the protection against H_2_O_2-_induced oxidative stress was considerably higher in the case of deacylhericenone. These findings suggest that the stronger bioactive form of the hericenone C compound is in fact deacylhericenone.

## 1. Introduction

*Hericium erinaceus*, commonly known as Yamabushitake in Japan, is a medicinal mushroom species attributed with several health benefits. This mushroom species has long been used in traditional Chinese medicine, and is known to have positive effects in health, including neuroprotection, and improvements in the digestive, circulatory, and immune systems [1]. Over the years, many studies have been dedicated to identifying the bioactive compounds in *H. erinaceus* [2,3,4]. Hericerins, erinacines, erinacerins, erinaceolactones, glycoproteins, polysaccharides, sterols, and various volatile compounds are some groups of bioactive compounds isolated from their fruiting bodies and mycelia [5]. A major characteristic compound in *H. erinaceus* belongs to a class of compounds known as hericenones, a meroterpenoid represented by its unique geranyl-resorcinol structure [6]. Hericenone C, one such meroterpenoid produced in the fruiting bodies of this mushroom, is known to be an important stimulant to produce nerve growth factors (NGF) in brain astrocytes [7]. Neurotrophic factors such as NGF and brain derived neurotrophic factor (BDNF) are protein molecules that perform nerve elongation, protection, and neogenesis [8,9]. The enhanced production of these proteins bears important implications for the improvement of neurodegenerative diseases such as dementia and Alzheimer’s disease [10].

Several in vivo studies involving animal models as well as human subjects have shown that *H. erinaceus* extracts improved cognitive functions and helped alleviate depressive disorders [11,12]. Oral supplements of *H. erinaceus* extracts containing known amounts of erinacine A, hericenone C, hericenone D, and L-ergothioneine in frail aged mice showed improvements in recognition memory, hippocampal and cerebellar neurogenesis, and locomotor performance [13,14]. The neuroprotective activities of hericenone C have been explored in several in vitro studies. It has been reported to stimulate the production of NGF in mouse astroglial cells [7] and promote neurite outgrowth in PC 12 cells when treated in conjunction with NGF [15]. However, another study reported that the production of NGF in human astroma cells was stimulated by the crude extract of *H. erinaceus* but not by hericenone C in its pure form [16]. 

Hericenone C consists of a fatty acid side chain linked by an ester bond, which is prone to in vivo enzymatic hydrolysis by ester hydrolases. Therefore, to test the hypothesis that the neuroprotective activity of hericenone C is the result of the hydrolyzed product rather than the intact compound, it was treated with pancreatic lipase to cleave the fatty acid side chain; and the plausible in vivo bioactive form of hericenone C was isolated and investigated for its bioactive effects. The lipase-derived compound (named deacylhericenone) was found to have higher BDNF mRNA transcription in SH-SY5Y and Caco-2 cells, and increased cell viability in oxidative stress-induced 1321N1 cells bioassay, compared to its parent compound. The results of this study provide strong evidence that the neuroprotective activity of hericenone C must, in fact, be attributed to its derivative, deacylhericenone. 

## 2. Results

### 2.1. Lipase Treatment of Hericenone C and n-Hexane Fraction

Hericenone C was treated with the lipase enzyme to cleave the fatty acid side chain from its core structure. As shown in Figure 1, compared to the total ion chromatogram (TIC) of hericenone C (Figure 1a), a new product peak was observed in the TIC chromatogram of the reaction solutions of hericenone C treated with lipase for 2 h (Figure 1c) and 24 h (Figure 1d). The peak abundance of hericenone C was dramatically decreased, and at the 24 h reaction time point, hericenone C was almost completely digested by lipase. The hericenone C derivative obtained after lipase treatment was isolated and analyzed by LC-QTOF-MS, and ^1^H-NMR. The deprotonated molecular ion [M-H]^−^ ion peak with *m/z* 331.1555 (calc. for C_19_H_23_O_5_ 331.1545) corresponded to a molecular formula of C_19_H_24_O_5_. It was speculated to be a structure resulting after the fragmentation of the fatty acid side chain (*m*/*z* 238) from hericenone C ([M-H]^−^ at *m*/*z* 569). Therefore, the deacylated derivative was named as deacylhericenone in this study. The ^1^H-NMR chemical shifts of deacylhericenone also supports the absence of the fatty acid side chain. The structures of the two compounds, hericenone C and deacylhericenone, are shown in Figure 2, with the hydrolysis process illustrated in Figure 3. The ^1^H and ^13^C-NMR spectra of compound **1**, ^1^H-NMR spectrum of compound **2**, and the HR-ESI-MS spectra of compound **1** and **2** are provided as Appendix A.

Compound **1** (Hericenone C): HR-ESI-MS 569.39047 [M-H]^−^, ^1^H-NMR (CDCl_3_, 400 MHz), aromatic moiety: δ_H_ 12.33 (s, HO-3), 6.51 (s, H-6), 5.31 (s, H-7), 10.09 (s, H-8), 3.90 (s, MeO-5), 3.37 (d, J = 7.28 Hz, H-1′), 5.31 (m, H-2′), 3.00 (br s, H-4′), 6.09 (s, H-6′), 1.82 (d, J = 1.29 Hz, H-8′), 1.76 (d, J = 1.32 Hz, H-9′), 2.11 (d, J = 1.29 Hz, H-10′), fatty acid moiety: δ_H_ 2.33 (dd, J = 7.71, 7.48 Hz), 1.58 (m), 1.23 (m), 0.866 (t, J = 7.04 Hz). ^13^C-NMR (150 MHz, CDCl_3_), aromatic moiety: δ_C_ 199.6 (C-5′), 193.1 (C-8), 163.5 (C-5), 163.0 (C-3), 155.5 (C-7′), 138.7 (C-1), 130.4 (C-3′), 126.3 (C-2′), 122.9 (C-6′), 117.3 (C-4), 112.9 (C-2), 105.6 (C-6), 62.9 (C-7), 56.0 (MeO-5), 55.6 (C-4′), 27.7 (C-8′), 21.6 (C-1′), 20.7 (C-10′), 16.4 (C-9′), fatty acid moiety; 173.2 (ester carbonyl), 34.3, 31.9, 29.72, 29.69, 29.61, 29.41, 29.39, 29.35, 29.25, 29.14, 29.11, 24.90, 22.72, 14.2. The assignment was confirmed through a comparison of the spectral data with those reported in the literature [7].

Compound **2** (Deacylhericenone): HR-ESI-MS 331.16545 [M-H]^−^, ^1^H-NMR (CDCl_3_, 400 MHz), δ_H_ 12.39 (s, HO-3), 6.52 (s, H-6), 4.95 (s, H-7), 10.20 (s, H-8), 3.90 (s, MeO-5), 3.37 (d, J = 7.42 Hz, H-1′), 4.93 (m, H-2′), 2.99 (br s, H-4′), 6.08 (s, H-6′), 1.83 (d, J = 1.22 Hz, H-8′), 1.76 (d, J = 1.31 Hz, H-9′), 2.11 (d, J = 1.22 Hz, H-10′).

### 2.2. Effect on BDNF mRNA Transcription

The effect of the ethanol extract of *H. erinaceus*, hericenone C, and deacylhericenone on the BDNF mRNA expression levels in 1321N1, SH-SY5Y, and Caco-2 cell lines can be seen in Figure 4. BDNF is primarily responsible for synaptic efficacy, and its expression has been found to be correlated with cognitive functions [17]. Therefore, the stimulation of expression levels can be helpful in discovering therapeutic compounds for improvement of neurodegenerative conditions. The mRNA expression was increased by almost 2 folds compared to control for all three cell lines in the case of ethanol extract of *H. erinaceus* at 100 µg/mL. The comparative expression levels for hericenone C and deacylhericenone showed that both compounds were able to significantly elevate expression levels in Caco-2, where deacylhericenone exhibited 3 folds higher levels compared to control. The gastrointestinal tract is innervated by vagal afferent nerves [18], which can transport neurotrophins expressed by intestinal epithelial cells. BDNF produced in the intestine is crucial for the growth and regeneration of enteric neurons, and the stimulation of the enteric circuit, intestinal peristalsis, and propulsion [19]. For the SH-SY5Y cell line, only deacylhericenone was found to significantly increase expression levels; whereas both the compounds were unable to have a prominent effect on mRNA expression in 1321N1 cells. 

### 2.3. Hydrogen Peroxide-Induced Oxidative Stress

The protective effect of the ethanol extract, hericenone C, and deacylhericenone against oxidative stress induced by H_2_O_2_ was evaluated in 1321N1 cells (Figure 5). In this study, an uninhibited H_2_O_2_-induced oxidative stress showed that the cell viability decreased to 19.9%, compared to the stress-free cells. Ethanol extract of *H. erinaceus* showed greater cell viability at 5 µg/mL (72.6%) than at a higher concentration of 10 µg/mL (59.9%), indicating the presence of some cytotoxic components in the extract. Hericenone C was unable to show significant protection against oxidative stress at the tested concentrations (1.6 to 12.5 µg/ mL). On the contrary, deacylhericenone showed a dose-dependent mitigation of oxidative stress, as indicated by the increasing cell viability with increasing concentration of the compound (1.6 to 12.5 µg/mL). At the highest tested concentration of 12.5 µg/mL, the cell viability was maintained up to 78.4%. From these findings, it can be deduced that the bioactivity of hericenone C is increased when the fatty acid side chain is cleaved off from hericenone C. 

## 3. Discussion

Several studies involving animal models and human subjects suggest that bioactive compounds in *H. erinaceus* have neuroprotective activities [14,20,21,22]. Therefore, extracts and isolated compounds from this mushroom species attract a lot of interest in terms of restorative and preventative care for neurodegenerative diseases. Cognitive functions were found to be improved in elderly human subjects given *H. erinaceus* supplements containing dried powder of the fruiting bodies [20,23]. An important consideration while interpreting the health benefits of bioactive compounds is their bioavailability upon consumption or administration into the human body. Compounds such as hericenone C, which have a fatty acid side chain linked by an ester bond to its core structure, are prone to enzymatic hydrolysis by enzymes such as pancreatic lipase. In such a scenario it becomes extremely important to determine whether the bioactive property is a result of the enzyme-hydrolyzed derivative or the parent compound. Upon treatment of the hericenone C solution with the pancreatic lipase, it was apparent that the hericenone peak in the TIC chromatogram significantly decreased over the 2 and 24 h time points. A simultaneous increase in the abundance of a new peak in Figure 1c,d suggested that a new product was formed because of the enzymatic hydrolysis of hericenone C. The new peak was confirmed to be deacylhericenone by LC-QTOF-MS, and ^1^H-NMR. 

To confirm that the bioactive properties are retained or even enhanced in deacylhericenone, a comparative bioassay was performed to test the neuroprotective activities of the deacylated product, the parent compound hericenone C, and the crude ethanolic extract of *H. erinaceus* fruiting bodies. The expression levels of the neurotrophic factor, BDNF has important implications in memory and learning, and therefore the cognitive function [24]. In vitro bioassays for the mRNA transcription of BDNF were performed in three cell lines known to produce this neurotrophic factor: 1321N1, SH-SY5Y, and Caco-2 cells. Our results showed that deacylhericenone has a stronger influence in the mRNA expression of neurotrophins, such as BDNF in the intestinal and some neuronal cells; and the total extract of *H. erinaceus* has a moderate but a broader influence, compared to the pure compounds. 

Most in vivo studies on the neuroprotective activities of *H. erinaceus* focus on the crude extracts of the fruiting bodies and mycelia [25]. While it has been well established that crude extracts have positive neuroprotective effects, in vivo studies focusing on individual bioactive compounds are limited. A previous report suggested that crude extracts of *H. erinaceus* and a pure compound hericene A markedly increased BDNF levels in mouse brain [26]. In another study, although crude extracts of *H. erinaceus* were able to enhance the expression of NGF in 1321 human astroma cells, hericenone C, D, and E were unable to promote NGF expression [16]. These findings are in consensus with the present study where BDNF mRNA transcription was enhanced in 1321N1 cells by the ethanolic extract of *H. erinaceus* but not by the pure compounds. However, significantly higher expressions were observed in SH-SY5Y and Caco-2 cell lines treated with deacylhericenone. Further studies must be done to elucidate the mechanism of action by which this compound delivers its neuroprotective actions.

Oxidative stress is one of the known causes of neurodegenerative diseases. The exposure to H_2_O_2_ overwhelms the innate antioxidant system of the cells, leading to cell injury or death, and ultimately to neurodegeneration [27]. Therefore, the ethanolic extract, hericenone C, and deacylhericenone were tested for their protective effect against oxidative stress induced by H_2_O_2_. Ethanolic extracts of *H. erinaceus* were found to protect H22 neurons from H_2_O_2_-induced oxidative damage [28]. Most studies investigating the effect of *H. erinaceus* on oxidative stress focus on the crude extracts rather than the isolated compounds [21,29]. The results of this study also showed that the ethanolic extract of *H. erinaceus* showed better protection against H_2_O_2_-induced oxidative stress in 1321N1 cells than the isolated compound hericenone C. However, it is important to note that the lipase-digested derivative, deacylhericenone, which is potentially the biotransformed form of the compound, showed a much greater protective effect against oxidative stress. 

The findings of the present study suggest that the in vivo bioactive form of the compound hericenone C is a deacylated derivative of the parent compound without its fatty acid side chain, which is named as deacylhericenone. Deacylhericenone was found to have higher neuroprotective abilities, as shown by greater stimulation of BDNF mRNA expression and oxidative stress reduction. Various mono and sesquiterpenoids have been reported to exhibit neuroprotective effects, but this is the first such report for the main meroterpenoid skeleton structure ensuing from the lipase decomposition of hericenone C. Further studies on the mechanism of neuroprotective actions and the bioavailability of deacylhericenone under in vivo conditions will help to confirm the efficacy of the compound for potential use as a nootropic agent.

## 4. Materials and Methods

### 4.1. Sample Extraction

The fruiting bodies of *H. erinaceus* mushrooms were obtained from Aso Biotech Co. (Kumamoto, Japan). The freeze-dried powder (83 g) of the mushroom was extracted in ethanol at 180 rpm for 24 h at room temperature. The extract was then filtered, and rotary evaporated at 45 °C under reduced pressure. The target compound, hericenone C, was isolated from the *n*-hexane soluble fraction of the ethanol extract and identified according to previously described methods [30].

### 4.2. Lipase Treatment of Hericenone C and n-Hexane Fraction

Hericenone C was subjected to lipase enzyme treatment. A 2 mg/100 µL acetone solution of hericenone C was treated with 50 U/mL lipase enzyme (Lipase AYS Amano, FJ353-46101, FUJIFILM Wako Chemicals) in 0.1 mol/L sodium phosphate buffer (pH 7.0) solution (Osaka, Japan). The mixture was incubated at 37 °C for 24 h. The reaction solutions were sampled at the time points of 0, 2, and 24 h. The samples were extracted using n-hexane: ethyl acetate (1:1), concentrated to dryness, then redissolved in methanol, and analyzed using LC-QTOF/MS (described in 4.4). The peak for the hydrolyzed product was observed in the samples collected at 2 and 24 h, respectively. 

The subsequent hericenone C deacylated derivative was isolated by scaling up the aforementioned enzymatic hydrolysis reaction method using 160 mg of n-hexane fraction (mainly containing hericenone C), followed by reverse phase preparative HPLC using Inertsil ODS-EP column (20 × 250 mm, 5 µm particle size) with a flow rate of 8 mL/min of 100% methanol, and UV detection at 239 and 290 nm. The structure of the isolated compound was then elucidated by ^1^H-NMR and LC-QTOF-MS analyses.

### 4.3. LC-QTOF-MS Analysis

The LC-QTOF-MS analysis was performed on an Agilent 1290 series UPLC system, equipped with a 1290 photodiode array detector (DAD) (Agilent Technologies, Santa Clara, CA, USA) coupled to an Agilent 6545 quadrupole-time of flight mass spectrometer (QTOF-MS) with a dual electrospray ionization (ESI) source to analyze Hericenone C and its hydrolyzed derivative. Each compound was dissolved in methanol (LC/MS grade, FUJIFILM Wako Pure Chemical Corporation) to a solution of 20 ppm and filtered with the 0.2 µm filter. A 1 µL portion of the solution was injected at a flow rate of 0.3 mL/min into the Agilent Poroshell 120 EC-C18 (2.1 × 100 mm, 2.7 µm) column. The column temperature was set at 40 °C and the peaks were also monitored using a UV detector at 295 nm beside the QTOF-MS detector. The mobile phase consisted of 0.1% formic acid in water (solvent A) and 0.1% formic acid in acetonitrile (solvent B). The mobile phase gradient program was set as B: 40% (0 min)—B: 99% (8 min)—B: 99% (14 min)—B: 40% (14.5 min)—B: 40% (19 min).The MS condition was electrospray ionization (ESI), negative ion polarity, gas temperature: 240 °C, drying gas flow: 12 L/min, nebulizer: 50 psi, sheath gas temperature: 350 °C, sheath gas flow: 11, fragmentor: 150 V, and VCap: 3000.

### 4.4. NMR Analysis

The ^1^H-nuclear magnetic resonance (^1^H-NMR) spectra were obtained with a JNM-ECS400 spectrometer (JEOL Ltd., Tokyo, Japan). ^13^C-Nuclear magnetic resonance (^13^C-NMR) spectra were recorded on a DRX-600 spectrometer (Bruker Daltonics, MA, USA). Chemical shifts are given as δ values relative to TMS as an internal standard, and coupling constants are given in Hz. The compounds were dissolved in Chloroform-d (99.8%, FUJIFILM Wako Pure Chemical Corporation, Osaka, Japan).

### 4.5. Cell Culture

Human astrocytoma cells (1321N1), human neuroblastoma cells (SH-SY5Y), and human colon cancer cells (Caco-2) were purchased from Riken Bioresource Center (Ibaraki, Japan). The 1321N1 cells were cultured in Dulbecco’s modified Eagle’s medium (DMEM, Gibco) supplemented with 5% fetal bovine serum (FBS), and 0.5% streptomycin/penicillin antibiotic solution. The SH-SY5Y and Caco-2 cells were cultured in DMEM supplemented with 10% FBS, 1% streptomycin/ penicillin antibiotic solution, and 1% non-essential amino acid (NEAA). All the cell lines were maintained in a 5% CO_2_ incubator at 37 °C.

### 4.6. Effect on BDNF mRNA Transcription

The effect of the ethanol extract of *H. erinaceus*, hericenone C, and deacylhericenone on the mRNA transcription of BDNF and NGF was tested in three cell lines known to produce these neurotrophic factors: 1321N1, SH-SY5Y, and Caco-2 cells [31]. 1321N1 and Caco-2 cells were seeded at the density of 2.0 × 10^5^ cells/well, and SH-SY5Y cells were seeded at the density of 4.0 × 10^5^ cells/well in 12 well plates. For 1321N1 cells, the medium was replaced with FBS free medium after 48 h incubation, and the sample was added after 24 h. The medium was replaced after 72 and 48 h for SH-SY5Y cells and Caco-2 cells, respectively, immediately followed by sample addition. After sample treatment for 3 (1321N1 cells) or 24 h (Caco-2 cells), the mRNA transcription level was evaluated using real-time PCR.

Total RNA was extracted using ISOGEN II (Nippon Gene), and the cDNA was synthesized using ReverTra Ace^®^ qPCR RT Master Mix with gDNA Remover (Toyobo). KAPA SYBR Fast qPCR kit (Kapa Biosystems) was used to perform quantitative real-time PCR. The manufacturer’s instructions were followed for all the kits used. Specific primers were used as follows: BDNF 5-GTCAAGTTGGGAGCCTGAAATAGTG-3 (forward), 5-AGGATGCTGGTCCAAGTGGTG-3 (reverse); NGF 5-CAACAGGACTCACAGGAGCA-3 (forward), 5-CTCTCCCAACACCATCACCT-3 (reverse); and β-actin 5-TCATGAAGTGTGACGTGGACATC-3 (forward), 5-CAGGAGGAGCAATGATCTTGATCT-3 (reverse). Gene expression relative to the housekeeping gene, β-actin, was calculated using the comparative threshold method (2^−ΔΔCt^).

### 4.7. Hydrogen Peroxide-Induced Oxidative Stress

The H_2_O_2_-induced oxidative stress was evaluated based on the previously described method [31]. The 1321N1 cells were seeded at the concentration 3 × 10^4^ cells/well in a 96-well plate. After 24 h, the cells were treated for 3 h with the sample, or the sample solvent in the case controls. Negative control (NC) is cells without H_2_O_2_ treatment, and test control (TC) is cells uninhibitedly treated with H_2_O_2_. Oxidative stress was induced by replacing the medium with fresh medium containing 100 µM H_2_O_2_. The replacement medium for NC did not contain H_2_O_2._ The cells were then incubated for 24 h, and the cell viability was then tested using MTT assay [32]. The cell survival was expressed as cell viability percentage with respect to NC. 

## 5. Patents

Kuniyoshi Shimizu. Brain-derived neurotrophic factor production promoters, nerve growth factor production promoters, oxidative stress inhibitors and their use. JP2021017420 A, 2021-02-15.

## Figures and Tables

**Figure 1 molecules-28-04549-f001:**
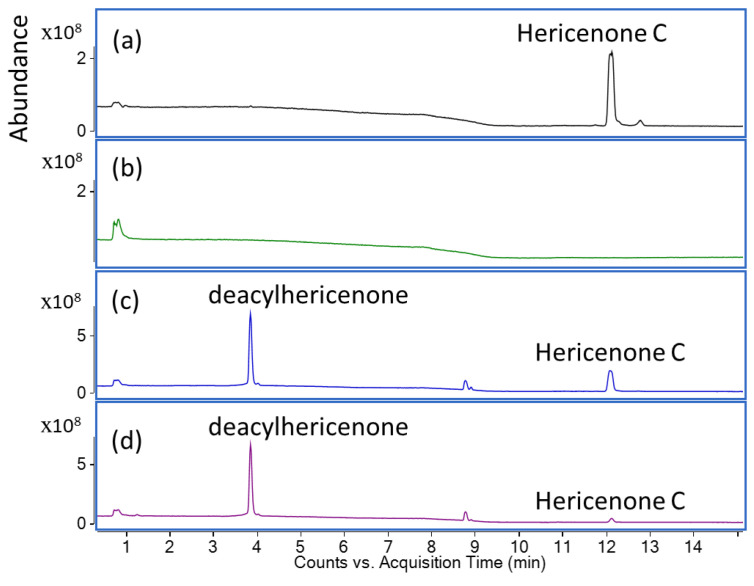
Total ion chromatogram (TIC) of hericenone C (**a**), lipase blank solution (**b**), the reaction solution of hericenone C treated with lipase for 2 (**c**) and 24 h (**d**).

**Figure 2 molecules-28-04549-f002:**
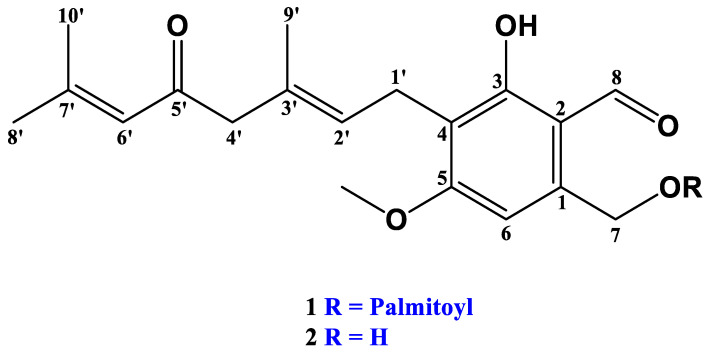
Structures of hericenone C (**1**) and its deacylated product (**2**).

**Figure 3 molecules-28-04549-f003:**
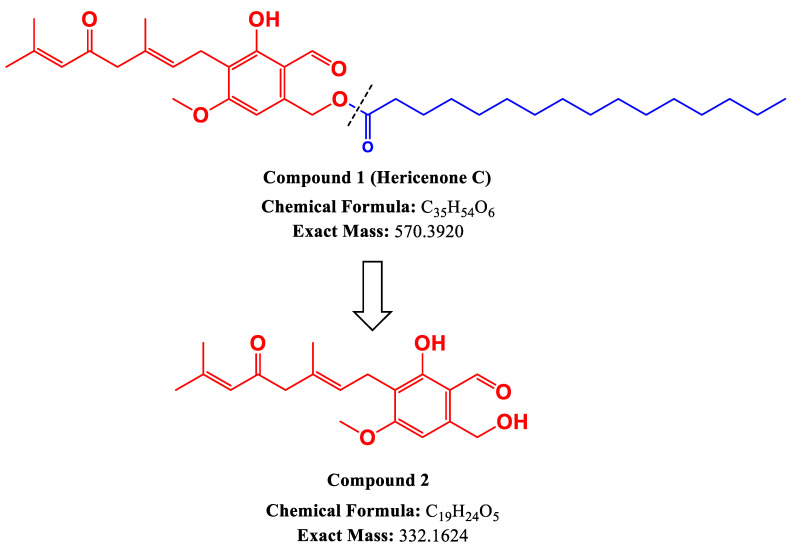
Hydrolysis of Hericenone C (**1**) to give the deacylated product (**2**).

**Figure 4 molecules-28-04549-f004:**
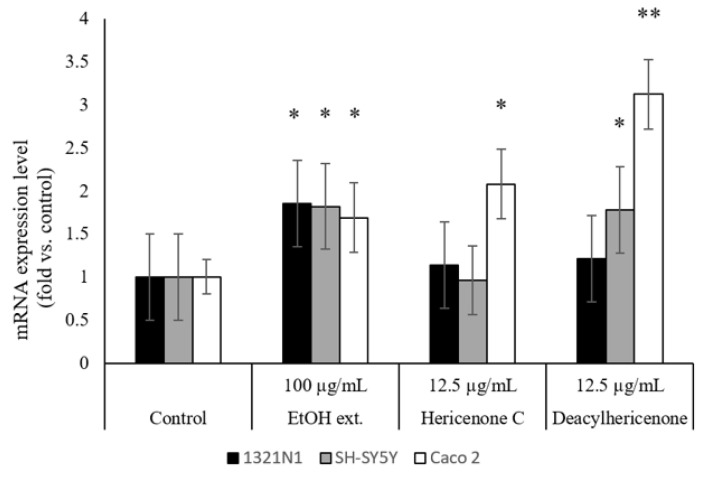
Effect of ethanol extract of *H. erinaceus*, hericenone C, and its derivative on the transcription of BDNF in 1321N1, SH-SY5Y cells, and Caco-2 cells. Significant difference between samples was analyzed by a Student′s *t*-test (** *p* < 0.01, * *p* < 0.1), *n* = 3.

**Figure 5 molecules-28-04549-f005:**
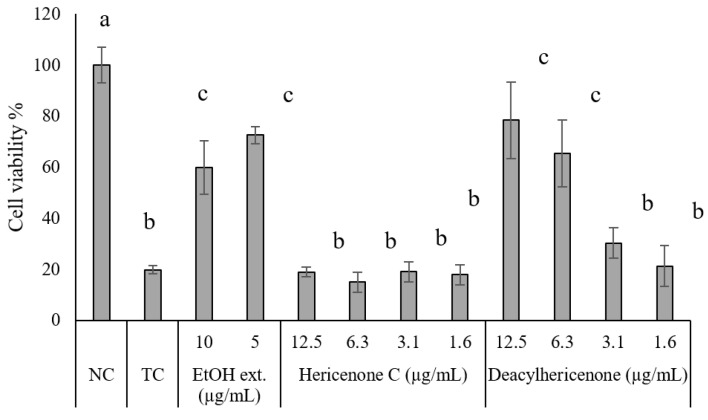
Effect of hericenone C and its decomposition product on oxidative stress toxicity mitigation in 1321N1 cells. Values are expressed as average ± SD, *n* = 4. A significant difference between samples was analyzed by a Tukey test (*p* < 0.01), *n* = 4. “NC” means H_2_O_2_ was not added to the cells, and “TC” means the cells were treated with H_2_O_2_ without inhibition. Different letters, a, b, and c, on top of the bars indicates a significant difference in cell viability between treatment groups compared to NC and TC.

## Data Availability

Data will be made available upon request.

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
