# Peer review of "Deacylated Derivative of Hericenone C Treated by Lipase Shows Enhanced Neuroprotective Properties Compared to Its Parent Compound"

_molecules, 2023, doi:10.3390/molecules28114549_

Round 1

Reviewer 1 Report

Comments to the Authors

The manuscript by Tamrakar et al. [Deacylated derivative of hericenone C treated by lipase shows enhanced neuroprotective properties compared to its parent compound] aimed to investigate the effects of lipase treatment on hericenone C, a bioactive compound found in Hericium erinaceus (Yamabushitake) mushroom, and evaluate the neuroprotective properties of its derivative, deacylhericenone.

 L22: Instead of saying "the resultant compound," specify that deacylhericenone is the compound obtained after lipase enzyme treatment.

L29-30: It is generally recommended to avoid using the same words in both the title and the keywords section. Modify.

L49: Instead of saying that hericenone C "has been purported to have various neuroprotective bioactivities," you should (based on previous research) mention specific neuroprotective effects that have been attributed to hericenone C. I suggest expanding the introduction part by discussing relevant in vivo studies that have been conducted so far. Furthermore, mentioning weaknesses or limitations of the existing studies would help highlight the importance of the submitted manuscript.

L134: H. erinaceus should be in italic form.

L134-137: something is wrong with the sentence. Missing dot?

Discussion part:

Give context: When mentioning previous studies, it would be beneficial to briefly summarize their findings and discuss how they relate to your study. Are there any connections between in vitro and in vivo findings?

Discuss limitations: Acknowledge the limitations of the study.

Broader implications & future directions: Discuss the potential implications of the findings in the broader field of neurodegenerative diseases and therapeutic interventions.

Since I am not a chemist, for an accurate review of the chemistry-related content in your text, I recommend the editor to consult with a chemist that will provide valuable feedback.

Minor editing of English language required.

Author Response

Response to Reviewer

Thank you very much for your comments and recommendations. All the changes in the main text have been indicated using red font. Here are our point-to-point responses and answers.

 L22: Instead of saying "the resultant compound," specify that deacylhericenone is the compound obtained after lipase enzyme treatment.

  • Thank you for the comment. The phrase “the resultant compound” has been removed and replaced with the following sentence in L22 to L23 of the revised manuscript:

“The compound formed after the lipase enzyme digestion was isolated and identified using LC-QTOF-MS combined with 1H-NMR analysis.”

L29-30: It is generally recommended to avoid using the same words in both the title and the keywords section. Modify.

  • The keywords repeated from the title have been removed and replaced with other keywords in L29.

L49: Instead of saying that hericenone C "has been purported to have various neuroprotective bioactivities," you should (based on previous research) mention specific neuroprotective effects that have been attributed to hericenone C. I suggest expanding the introduction part by discussing relevant in vivo studies that have been conducted so far. Furthermore, mentioning weaknesses or limitations of the existing studies would help highlight the importance of the submitted manuscript.

  • Thank you for the suggestion. Relevant discussion on previous research has been added to expand the introduction part from L49 to L59 of the revised manuscript.

L134: H. erinaceus should be in italic form.

  • ericaneus has been italicized throughout the revised manuscript.

L134-137: something is wrong with the sentence. Missing dot?

  • The missing period has been added to separate the sentences in L144 to L146 in the revised manuscript.

Discussion part:

Give context: When mentioning previous studies, it would be beneficial to briefly summarize their findings and discuss how they relate to your study. Are there any connections between in vitro and in vivo findings?

  • Relevant discussions on the in vitro and in vivo findings and their relevance to the present study have been added to the discussion part. Since most in vivo studies focus on crude extracts of Hericium erinaceus, it is difficult to make a direct comparison between the in vivo studies and the in vitro studies for the isolated compounds hericenone C and deacylhericenone.

Discuss limitations: Acknowledge the limitations of the study.

Broader implications & future directions: Discuss the potential implications of the findings in the broader field of neurodegenerative diseases and therapeutic interventions.

  • Thank you for your suggestions. The limitation of the study and the need for further investigations, as well as the broader implications of the study for the future have been added to the discussion part.

Reviewer 2 Report

The investigation through fundamental experimental research of some potential compounds responsible for the neuroprotective effect of Hericium erinaceus is an approach of major scientific importance.

The authors developed this manuscript professionally; they used eloquent investigative techniques. The chapters and subchapters contain relevant information and are accompanied by clear images. The results are presented fluently, based on logical arguments. Instrumental investigations and genetic analyzes were addressed and discussed in detail. In addition, the authors proved a good knowledge of some previous research carried out by other researchers in similar issues. It is remarkable that the original findings have already been patented.

A typo should be corrected: Line 187, please replace again with against.

Author Response

Response to Reviewer:

Thank you very much for your comments and recommendations. All the changes in the main text have been indicated using red font. Here are our point-to-point responses and answers.

A typo should be corrected: Line 187, please replace again with against.

  • Thank you for pointing out the typo. The word “again” has been replaced with “against” in Line 206 of the revised manuscript.

Reviewer 3 Report

The idea of the paper is very interesting and the paper is well written but please consider these minor comments:

1- The name of the plant should be written italic in the whole manuscript.

2- Write the complete letters of the abbreviated words one time in the manuscript, then write the abbreviation

3- I suggest that the sequence of the primers to be transferred into the supplementary file.

Author Response

Response to Reviewer

Thank you very much for your comments and recommendations. All the changes in the main text have been indicated using red font. Here are our point-to-point responses and answers.

  • The name of the plant should be written italic in the whole manuscript.
  • Thank you for your comment. Hericium erinaceus has been italicized throughout the revised manuscript.
  • Write the complete letters of the abbreviated words one time in the manuscript, then write the abbreviation.
  • Thank you for pointing this out. The relevant changes have been made to the revised manuscript.
  • I suggest that the sequence of the primers to be transferred into the supplementary file.
  • Thank you for your suggestion. We believe that the sequence of primers is an important information for readers interested in the mRNA transcription experiment. Therefore, if possible, we could like to retain this information in the main text.